# Distribution and New Records of the Bluntnose Sixgill Shark, *Hexanchus griseus* (Hexanchiformes: Hexanchidae), from the Tropical Southwestern Atlantic

**DOI:** 10.3390/ani13010091

**Published:** 2022-12-26

**Authors:** Jones Santander-Neto, Getulio Rincon, Bruno Jucá-Queiroz, Vanessa Paes da Cruz, Rosângela Lessa

**Affiliations:** 1Instituto Federal de Educação, Ciência e Tecnologia do Espírito Santo, Campus Piúma, Rua Augusto Costa de Oliveira, 660, Praia Doce, Piúma 29285-000, Brazil; 2UFMA Campus Pinheiro, Universidade Federal do Maranhão, Pinheiro 65200-000, Brazil; 3SOD Geotecnia Construções, Rua do Sol, 300, Aleixo, Manaus 69060-084, Brazil; 4Departamento de Biologia Estrutural e Funcional, Instituto de Biociências, Universidade Estadual Paulista Júlio de Mesquita Filho-UNESP, Botucatu 01049-010, Brazil; 5Departamento de Pesca e Aquicultura, Rua Dom Manoel de Medeiros, s/n, Dois Irmãos, Universidade Federal Rural de Pernambuco-UFRPE, Recife 52171-900, Brazil

**Keywords:** fisheries, deep sea, DNA barcoding, elasmobranchs, Chondrichthyes

## Abstract

**Simple Summary:**

The bluntnose sixgill shark, *Hexanchus griseus*, is a widely distributed species found in all oceans, even if irregularly, inhabiting continental shelves and slopes, islands, and mid-ocean ridges in deep seas. The species is currently classified by the International Union for Conservation of Nature (IUCN) as globally near-threatened, with a decreasing population. Despite some records in Brazil, the known distribution of this species in the Southwestern Atlantic is very patchy and, in some cases, not yet recorded in the world reference literature, such as in northeastern Brazil. This study, therefore, highlights new records for the species in the Tropical Southwestern Atlantic, particularly off the northeastern Brazilian coast. This information is paramount to establishing distribution maps for the species employed in a population status assessment by the IUCN.

**Abstract:**

The bluntnose sixgill shark, *Hexanchus griseus*, is a widely distributed demersal species found in tropical and temperate waters of the Pacific, Atlantic, and Indian Oceans, inhabiting continental shelves and slopes, islands, and mid-ocean ridges at depths ranging from 200 to 1100 m. In the Southwestern Atlantic, this species has been recorded from northeastern to southern Brazil, Argentina, and Uruguay. Despite this, the known distribution of this species in the Southwestern Atlantic is very patchy and, in some cases, still mostly ignored in the literature, such as in northeastern Brazil. This study, therefore, aimed to report 23 new records of *Hexanchus griseus* in the Tropical Southwestern Atlantic and highlight the presence of this species off the northeastern Brazilian coast. So far, *Hexanchus griseus* was officially reported from the Fernando de Noronha Archipelago, Saint Peter and Saint Paul Archipelago, and the state of Ceará along the northeast coast of Brazil. Herein, the known distribution is extended to the continental shelf breaks and upper slopes of other Brazilian states, reinforcing the previously reported occurrence of the species near the Saint Peter and Saint Paul Archipelago.

## 1. Introduction

Sixgill sharks belonging to the Hexanchidae Gray family, 1851, present unicuspidate front teeth in the upper jaw and comb-shaped and blade-like in the lower jaw, six or seven pairs of gill slits, and a fairly stocky body. The Hexanchidae family comprises three genera and five species. The bluntnose sixgill shark, *Hexanchus griseus* (Bonaterre, 1788), is a widely distributed demersal species found in tropical and temperate waters of the Pacific, Atlantic, and Indian Oceans [1], although with a patchy distribution, and can inhabit continental shelves and slopes, islands, and mid-ocean ridges at depths ranging from 200 to 1100 m, and able to extend down to at least 2500 m [2,3,4].

The bluntnose sixgill shark is a large-bodied shark ranging up to 550 cm in total length (TL), reaching maturity at about 310 cm and over 400 cm TL, for males and females, respectively [4,5]. Its reproductive cycle is still not well defined, but according to different studies [5,6,7,8], it is believed to be biannual, with a resting phase for females of 12 months followed by a gestation period of 12 months. This species is known to have very large litters of 47–108 pups born at around 55–74 cm TL [4,5,6,8]. A polyandrous mating system was detected in an analysis concerning the genetic relationship between a large female found on the beach and 71 individuals from a litter, with up to nine males contributing to her offspring, which confers a significant evolutionary success for the group [9]. This mating system can influence populations, leading to increased levels of genetic variability and decreasing inbreeding [10,11].

In the Southwestern Atlantic, bluntnose sixgill sharks have been recorded from northeastern to southern Brazil [12,13,14], Argentina, and Uruguay [15]. Despite this, the known distribution of this species in the Southwestern Atlantic is very patchy and, in some cases, still mostly ignored in the literature’s main references, such as in northeastern Brazil [3,4]. Due to their biology and increasing anthropogenic actions, the bluntnose sixgill shark is currently classified by the International Union for Conservation of Nature (IUCN) as globally near-threatened, with a decreasing population trend [3].

In this context, this study aims to compile all available data on the species occurrence in this region including new records from the Tropical Southwestern Atlantic, highlighting the presence of this species off the northeastern Brazilian coast.

## 2. Materials and Methods

Data on the occurrence of the *Hexanchus* genus in the Tropical Southwestern Atlantic were obtained from three different sources, namely artisanal fishing landing sites, bottom longline fisheries (commercial fisheries), and handline fishing using line reels.

All analyzed specimens were identified based on specific literature [1,4]. Morphometric measurements, when possible, were determined as the length in centimeters and weight in kilograms. When the specimens were not available, total length (TL) and total weight (TW) was estimated by those who had access to the animals and corroborated by photographs or videos. 

Care and use laws for experimental animal welfare were not applied in this study due to the nature of data collection from commercial fishing landings. The specimens were commercialized by fishers and we did not retain them for any other purposes. Even so, authorization from the Chico Mendes Institute for Biodiversity was obtained (ICMBio—SISBIO License n°49663-2). Concerning the sampling efforts conducted under the Program for Evaluation of Living Resources in the Exclusive Economic Zone (REVIZEE), a government program created to assess the potential for sustainable exploitation of all marine natural resources of Brazil’s exclusive economic zone, the program was finalized before the Biodiversity Authorization and Information System (SISBIO) development, in 2007.

### 2.1. Bottom Longlines

Data from six scientific expeditions conducted in northeastern Brazil between 1997 and 1999 (two per year) were obtained under the REVIZEE program. These cruises operated with bottom longlines containing 500 hooks, each 13/0 in size (Mustad hook). Fishing activities took place at different depths, from 50 to 500 m every 50 m.

Further data from 53 commercial fishing trips along the northeastern Brazilian coast were obtained from November 2004 to August 2011. These fisheries operated with bottom longlines, with hooks numbering between 250 and 900 and hook sizes varying from 13/0 to 16/0 (Mustad hooks), with the predominant use of the 15/0 and 16/0 sizes. The fishing activities took place near the 100–200 m isobaths from latitude 1°20′ S to 11°30′ S.

### 2.2. Handline Using Line Reels

A video was recorded during a fishing trip around the Saint Peter and Saint Paul Archipelago (ASPSP) (coordinates: 00°55′03” N/29°20′43″ W). After setting the longline, a handline using line reels was also prepared and set, from which one bluntnose sixgill shark specimen was caught. The line was approximately 200 m long employing 13/0 hooks.

### 2.3. Artisanal Fishing Landing Site

One bluntnose sixgill shark specimen was landed in an artisanal fishing landing site located in the Mucuripe Embayment, Fortaleza, in the state of Ceará, Brazil. The fishing area for this small-scale fishing fleet comprise 3217 km^2^ bordered at 03°43′S/038°05′W; 03°23′ S/038°05′ W and 03°25′S/038°48′ W; 03°01′S/038°49′ W (Figure 1). According to the fishmonger that sent photos and a video, the specimen was caught with a hook size number 12/0 at depths between 70 and 120 m. The carcass (without head, fins, guts, and skin) was sold to a fish butcher. A tissue sample was obtained from this specimen to proceed with the DNA barcoding analysis.

### 2.4. DNA Barcode Analysis

For more accurate identification, identification was performed based on barcode DNA, using a small piece of the muscle tissue (40 mg) preserved in ethanol 95%. DNA was extracted using the one silica-based extraction method, the standard Canadian Centre for DNA Barcoding (CCDB) DNA extraction protocol using a glass fiber plate [16] following the corresponding instructions. One thousand and nine hundred microlitres of a lysis buffer (Lysis Buffer or T1) containing Proteinase K (Life Technologies; Invitrogen, Carlsbad, NM, USA) was added to the sample that was then ground with a pestle inside the tube and incubated overnight at 56 °C. Two-hundred-microlitre aliquots of the lysate were then transferred into Ultident tube racks, mixed with 400 μL of corresponding binding buffer, and then transferred to a binding plate [Bioinert membrane Acroprep plate (Pall Life Sciences, Ann Arbor, MI, USA)], which was then centrifuged at 6000× *g*. The first wash volume consisted of 400 μL, followed by a second wash of 750 μL using the corresponding buffer systems. The plate was then incubated at 56 °C for 30 min, and DNA was eluted with 50 μL of H_2_O preheated to 56 °C.

A polymerase chain reaction (PCR) was performed using a previously published primer [17]. A Veriti 96 thermocycler (Applied Biosystems) was used following the protocol suggested by the Platinum^®^ Taq DNA Polymerase manual (Invitrogen). PCR amplicons were visualized on a 1% agarose gel E-Gels (Invitrogen). Fragments of approximately ~650 base pairs (bp) barcode region of the cytochrome C oxidase subunit I COI gene were bi-directionally sequenced using a BigDye Terminator v.3.1 Cycle Sequencing Kit (Applied Biosystems, Inc.) employing an ABI 3730 capillary sequencer following the manufacturer’s instructions.

Sequence files were assembled, and consensus sequences were constructed, aligned, and trimmed using the Geneious 4.8.5 software [18], and submitted to GenBank (Accession no. OP897150). The extended reference databases BOLD and GenBank were used, obtained through downloading sequences for *H. griseus, H. nakamurai,* and *H. vitulus*. The sequences were then aligned with the muscle algorithm [19] implemented in the Geneious 4.8 software. Interspecific and intraspecific distances were analyzed using the Kimura-2-parameter model (K2P) [20] available in the MEGA X software [21]. A maximum likelihood (ML) analysis was performed through the RAxML v8.2 software [22] with the GTR GAMMA model, and a posteriori bootstrap analysis was conducted employing the autoMRE function with Randomized Axelerated Maximum Likelihood (RAxML) criterion and the best-fitting ML tree was reconciled with the bootstrap replicates. The tree was viewed and edited in FigTree v1.4.3. software (http://tree.bio.ed.ac.uk/software/figtree/, accessed on 1 June 2022).

## 3. Results

From November 1998 to December 2020, 23 *Hexanchus griseus* specimens were recorded in the Tropical Southwestern Atlantic using bottom longlines, handlines using reels, and at an artisanal fishing landing site (Table 1).

### 3.1. Bottom Longlines

Two of six scientific expeditions during the REVIZEE program in 1998 provided the following eight records for *Hexanchus griseus* in northeastern Brazil (Table 1; Figure 1): Five specimens from Cabo Calcanhar, in the state of Rio Grande do Norte (RN) were caught in November 1998, ranging from 80 to 240 m in depths, one specimen captured at Baía Formosa, RN, in November 1998 at a depth of 250 m and two specimens captured at Pitimbu in the state of Paraíba, December 1998, ranging from 260 and 450 m in depth (Figure 2a,b). 

Twelve *Hexanchus griseus* specimens were caught in seven out of the fifty-three commercial bottom longline fishing trips on the seamounts of the Brazilian North Chain off the state of Ceará from 2010 to 2011. One additional specimen was captured from the external continental shelves of the state of Maranhão in 2005 (Table 1).

### 3.2. Handline Using Line Reels

On 29 March 2010, a female specimen with an estimated total length between 280 and 300 cm was caught at Lat 0°54.410′ N, Long 29°21.892′ W (off the Saint Peter and Saint Paul Archipelago) at a depth of 205 m and only recorded in a video (Appendix A) before the animal was released by the fishermen due to difficulties in bringing it aboard.

### 3.3. Artisanal Fishing Landing Site

One specimen was captured 48 nautical miles (NM) off the coast of the state of Ceará, Brazil, and landed on 13 December 2020. A fishmonger sent photos and a video (Figure 2c,d; Appendix A). According to him, the female specimen was approximately 180 cm TL. A barcode sequence with more than 600 bp was obtained from this specimen, in addition to 36 sequences obtained from BOLD and GenBank, resulting in a final matrix of 37 sequences (23 *H. griseus*, 12 *H. vitulus,* and two *H. nakamurai*) (Figure 3). Insertions, deletions, and stop codons were not observed. Following alignment and editing, the final matrix contained 605 characters, 212 of which were variable, comprising 29.1% adenine, 21.8% cytosine, 31.8% thymine, and 17.2% guanine. The genetic distance analysis clearly splits the three Hexanchus species, with 0.605 ± 0.021 distance between *H. griseus* and *H. vitulus*, 0.640 ± 0.011 between *H. vitulus* and *H. nakamurai*, and 0.077 ± 0.012 between *H. griseus* and *H. nakamurai*. These findings reveal a low intraspecific genetic variation of <0.001 in all species. The RAxML tree displayed long branches structuring each of the three *Hexanchus* species, with the clear separation of *H. griseus, H. nakamurai,* and *H. vitulus*, with more than 85% bootstrap support, confirming the captured specimen as *H. griseus*.

## 4. Discussion

*H. griseus’s* distribution in the Tropical Southwestern Atlantic is still not clear and presents some discrepancies in its literature-reported distribution where the species is widely reported for the region [12] or not reported at all [3,4]. Considering the records reported herein, the bluntnose sixgill shark seems to present a wide distribution range along the northeastern Brazilian continental slope, islands, and seamounts, rising to shallower waters in specific areas where specimens are caught from 70 to at least 450 m. The shelf-break demersal habitat of the species and the lack of specific deep-water fisheries, however, limit the understanding of the species distribution pattern [23] due to inherent difficulties in identifying and recording catches and landings [24]. The species is, thus, seldomly recorded.

The loose and oily muscles present in bluntnose sixgill sharks, typical of deep-water fishes, allied to the large size of these animals are the main reasons why fishermen avoid boarding them. Furthermore, fishermen indicate that the meat decomposes very fast and contaminates other fish, while also having no market value. It may, however, be used as bait when brought aboard, which makes it difficult to record captures.

The bluntnose sixgill shark has been reported at the Fernando de Noronha islands [25] and along the entire Brazilian Northeast coast [10] based on conference abstracts [26,27]. Considering the long extension of the Brazilian coast, we recovered the original data of these unreported captures and added new captures in order to provide a better interpretation of the species distribution. So far, *Hexanchus griseus* has been officially reported along the northeast coast of Brazil at the Fernando de Noronha Archipelago, the Saint Peter and Saint Paul Archipelago, and the state of Ceará [13,14,25].

With the data reported herein, the known distribution of the species is extended to the continental shelf breaks and upper slopes of the states of Maranhão, Rio Grande do Norte, and Paraíba. New reports are also provided for the Brazilian North Chain seamounts where twelve specimens were captured on different occasions. In addition, the present record of a female specimen captured near the Saint Peter and Saint Paul Archipelago (with the video recording provided herein as Appendix A) reinforces a previously reported occurrence of the species [14] in a biodiversity estimation study using baited remote underwater stereo-videos (stereo-BRUV’s). These recurrent catches and video recordings suggest resident populations along the Brazilian North Chain and the oceanic islands of Fernando de Noronha and the SPSP Archipelago. However, only a larger study on the population genetics of this species would define if these animals belong to multiple or a single widely distributed population. Moreover, relevant literature still does not consider the presence of the species in the region [3,4], which is concerning with regard to local and regional biodiversity evaluation statuses as well as the fisheries interaction and potential population impacts.

The bluntnose sixgill shark presents an almost global continuous distribution alongside continental and insular shelf breaks and slopes, seamounts, and even bays connected to submarine canyons and trenches associated with colder waters from upwelling systems [28,29,30], also recorded in shallow waters of Tasmanian rivers [31]. It is possible that *H. griseus*, like other deep-sea species, may present wider distributions as they live in waters with restricted temperature variations that extend to different latitudes and depths. Therefore, this species may be expected to also occur in other states in northeastern Brazil where its occurrence has not yet been reported. In Brazil, a more common occurrence of the species has been reported for the southern region of the country, in the states of Santa Catarina and Rio Grande do Sul [12,23,32], corroborated by large captures later in southern Brazil [33]. However, the present data has raised suspicions about the possibility that the species may occur more frequently in northeastern Brazil than previously expected, due to the lack of occurrence records. Furthermore, it is important to maintain sampling efforts and use as many available tools as possible to elucidate the diversity and distribution of this genus in the Atlantic Ocean [34].

The *H. griseus* specimens reported herein were found at depths ranging between 70 and 450 m, corroborating the known depth range for the species [3], with most catches taking place at depths up to 250 m, probably due to the effective fishing gear performance depth in the monitored fisheries. It is possible that fisheries prospecting in waters deeper than 200 m may also capture the bluntnose sixgill shark.

All sex-identified specimens were females with total lengths ranging from 180 to 300 cm, indicating that they had not yet reached sexual maturity [4,5], similar to the male previously recorded in the state of Ceará [13]. The absence of adult specimens in the catches is probably due to fishing gear damage (cable breakage, broken hook, and others) due to the large size of adults, or fishermen’s refusals to ship large specimens commercially devalued. This rarity concerning adult specimen records seems to be common worldwide [6,15,35]. However, it would be interesting to identify the species’ size and sexual structure (life stages) along its bathymetric distribution, to better understand how the species uses the region. A recent study indicated that the sexual distinction observed by Sr/Ca *H. griseus* ratios could indicate spatial segregation, with the possibility that these ratios influence physiological differences between sexes [36]. Research on the spatial ecology of the species has already been suggested [31,36] but is still lacking.

Considering that *H. griseus* occupies a prominent position in the regulation of multiple prey in deep environments [31], further knowledge regarding the distribution of this species would aid in understanding how it can contribute to environmental equilibrium. In addition, these data will also aid in the construction of maps on the species distribution, which is paramount in regional population assessments aiming at Chondrichthyes conservation.

## 5. Conclusions

The new records provided herein for *Hexanchus griseus* in the Tropical Southwestern Atlantic extend the known distribution of the species to the states of Maranhão, Rio Grande do Norte and Paraíba in Brazil.

## Figures and Tables

**Figure 1 animals-13-00091-f001:**
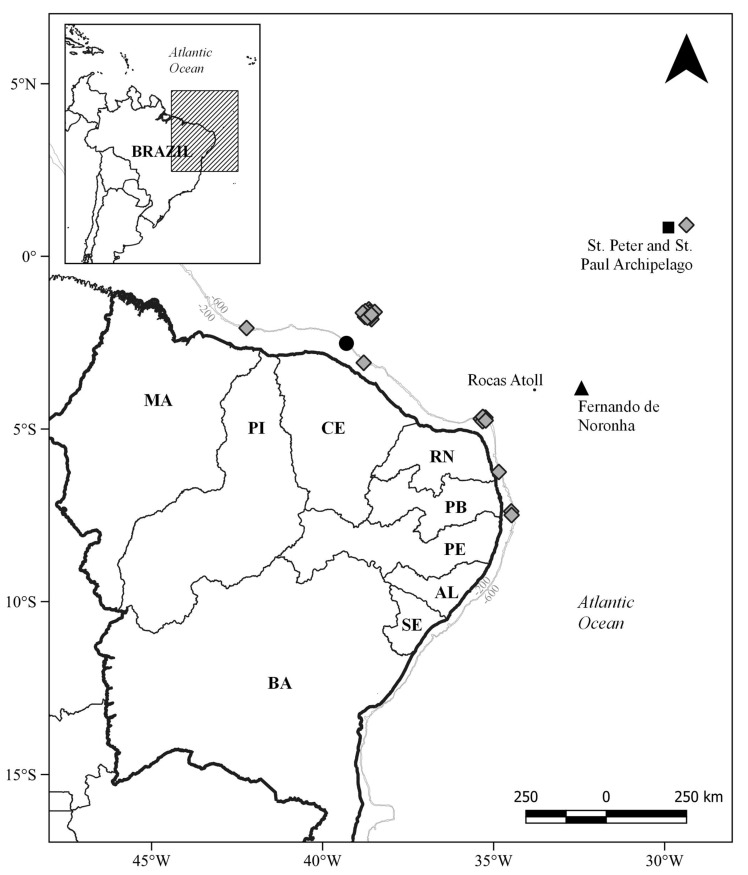
Map depicting previous and additional records for the bluntnose sixgill shark, *Hexanchus griseus*, from the Tropical Southwestern Atlantic. Gray diamonds indicate new records and black forms indicate previous records. Black triangle: Moreira-Junior (1993); black circle: Santander-Neto et al. (2007); black square: Pinheiro et al. (2020); MA: Maranhão; PI: Piauí; CE: Ceará; RN: Rio Grande do Norte; PB: Paraíba; PE: Pernambuco; AL: Alagoas; SE: Sergipe; BA: Bahia.

**Figure 2 animals-13-00091-f002:**
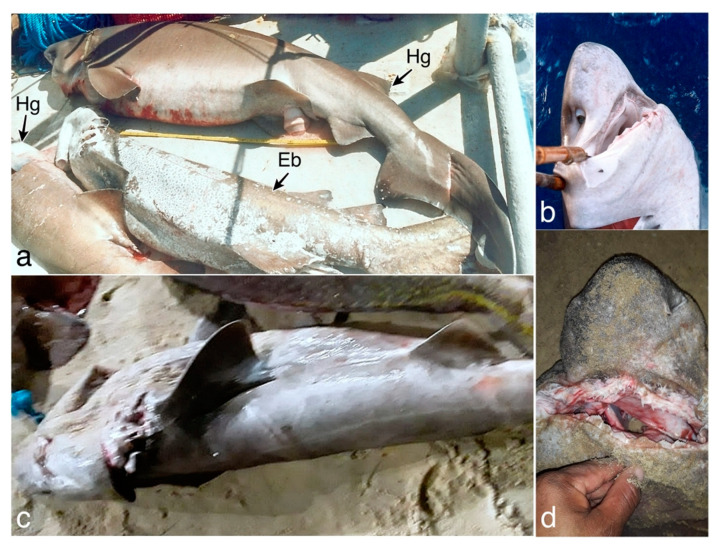
Female *Hexanchus griseus* specimen (**a**) caught off the state of Paraíba, Brazil, and (**b**) the specimen’s head during the landing and; and another female specimen (**c**) landed at the Mucuripe Embayment, in the state of Ceará, Brazil, photographed by the fishmonger with a (**d**) ventral view of the head. Hg, *Hexanchus griseus*; Eb, *Echinorhinus brucus*.

**Figure 3 animals-13-00091-f003:**
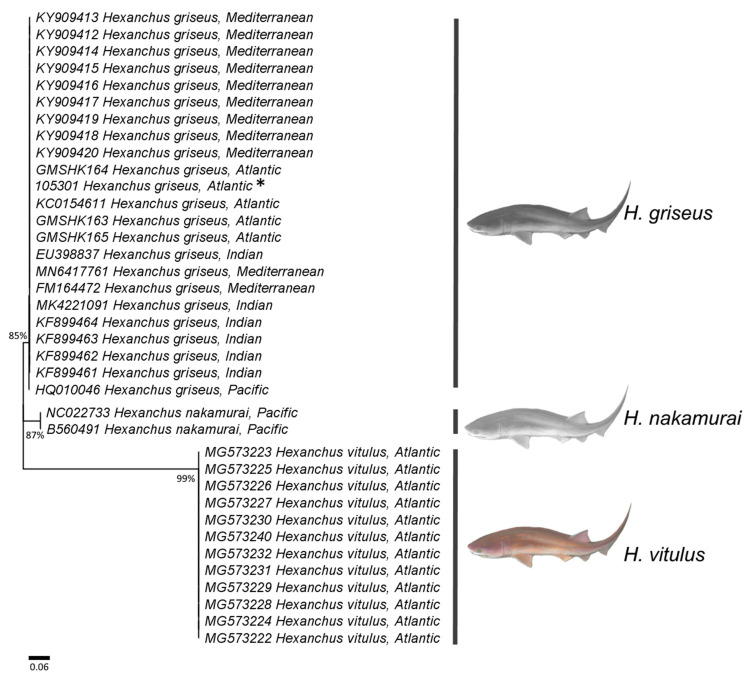
Relationships between *Hexanchus* spp. based on a maximum likelihood analysis and inferred from mitochondrial cytochrome c oxidase subunit I (COI) gene sequences. All nodes displayed bootstrap values above 85%. * indicates the sample collected in the present study.

**Table 1 animals-13-00091-t001:** Information on the *Hexanchus griseus* specimens recorded in the Tropical Southwestern Atlantic from 1998 to 2020. Bottom longline, BL; artisanal fishing landing site, AFLS; handline using line reels, HLR; Maranhão, MA; Ceará, CE; Rio Grande do Norte, RN; Paraíba, PB; Saint Peter and Saint Paul Archipelago, ASPSP. N, number of specimens; TW, total weight in kg; TL, Total length in cm.

Date	Source	Coordenadas	Location and State	N	Depth	TW	TL	Sex
17 November 1998	BL	04°43′01″ S/035°18′39″ W	Cabo Calcanhar, RN	1	240			
17 November 1998	BL	04°43′01″ S/035°18′39″ W	Cabo Calcanhar, RN	1	240			F
17 November 1998	BL	04°43′01″ S/035°18′39″ W	Cabo Calcanhar, RN	1	240			F
17 November 1998	BL	04°43′22″ S/035°20′18″ W	Cabo Calcanhar, RN	1	82			
17 November 1998	BL	04°43′31″ S/035°19′58″ W	Cabo Calcanhar, RN	1	81			F
29 November 1998	BL	06°14′34″ S/034°51′06″ W	Baía Formosa, RN	1	250	116	294	F
2 December 1998	BL	07°24′10″ S/034°27′39″ W	Pitimbu, PB	1	260	82	248	F
2 December 1998	BL	07°24′33″ S/034°27′08″ W	Pitimbu, PB	1	450	85.5	267	F
August 2005	BL	02°04′39″ S/042°13′32″ W	Tutóia, MA	1	100–200	35		
29 March 2010	HLR	00°54′24″ N/029°21′54″ W	ASPSP	1	205		~280–300	F
June 2010	BL	~01°40′ S/038°40′ W	Oceanic banks, CE	2	100–200	300 *		
August 2010	BL	~01°40′ S/038°40′ W	Oceanic banks, CE	2	100–200	100 *		
September 2010	BL	~01°40′ S/038°40′ W	Oceanic banks, CE	3	100–200	200 *		
September 2010	BL	~01°40′ S/038°40′ W	Oceanic banks, CE	1	100–200	100		
December 2010	BL	~01°40′ S/038°40′ W	Oceanic banks, CE	1	100–200	110		
July 2011	BL	~01°40′ S/038°40′ W	Oceanic banks, CE	3	100–200	200*		
13 December 2020	AFLS	~03°05′ S/038°48′ W	CE	1	70–120		~180	F

* combined total weight of the specimens in the fishery.

## Data Availability

Not applicable.

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
