# Peer review of "Distribution and New Records of the Bluntnose Sixgill Shark, Hexanchus griseus (Hexanchiformes: Hexanchidae), from the Tropical Southwestern Atlantic"

_animals, 2022, doi:10.3390/ani13010091_

Round 1

Reviewer 1 Report

Dear editors, authors,

It was my pleasure to read and review the paper by Santander-Neto et al., entitled ‘Distribution and new records of the Bluntnose Sixgill Shark Hexanchus griseus (Hexanchiformes: Hexanchidae), from Southwestern Tropical Atlantic’.

This topic (i.e., new records) is of interest, especially in view of the advances we need in the correct assessment of the distribution and the conservation status of this iconic and threatened shark species, as well as for future conservation measures.

The general structure of the paper is OK, but it is somewhat annoying that some sentences are repeated several times throughout the manuscript (i.e. in the simple summary, the abstract, the discussion, and the conclusions). I would suggest to delete some (e.g., conclusions needed?), or to make these parts more complement to each other. Also the English reads quite difficult in several occasions so that a check by a native English speaker is highly recommended.

In terms of presentation, I would suggest to put Figure 1 (map) before the Table 1. And please provide credits for the shark illustrations used in Figure 3.

Finally, please give some attention to the list of references that need several corrections and additions (missing names and locations of publishers, editors, pages, etc.). I would also suggest to include the two url’s of the youtube videos into the references, and not in the main text.

For several smaller comments, see the annotated manuscript attached.

I hope these suggestions will improve the quality of the paper.

Best regards,

(reviewer)

Author Response

Dear editors, authors,

It was my pleasure to read and review the paper by Santander-Neto et al., entitled ‘Distribution and new records of the Bluntnose Sixgill Shark Hexanchus griseus (Hexanchiformes: Hexanchidae), from Southwestern Tropical Atlantic’.

This topic (i.e., new records) is of interest, especially in view of the advances we need in the correct assessment of the distribution and the conservation status of this iconic and threatened shark species, as well as for future conservation measures.

The general structure of the paper is OK, but it is somewhat annoying that some sentences are repeated several times throughout the manuscript (i.e. in the simple summary, the abstract, the discussion, and the conclusions). I would suggest to delete some (e.g., conclusions needed?), or to make these parts more complement to each other. Also the English reads quite difficult in several occasions so that a check by a native English speaker is highly recommended.

In terms of presentation, I would suggest to put Figure 1 (map) before the Table 1. And please provide credits for the shark illustrations used in Figure 3.

Finally, please give some attention to the list of references that need several corrections and additions (missing names and locations of publishers, editors, pages, etc.). I would also suggest to include the two url’s of the youtube videos into the references, and not in the main text.

For several smaller comments, see the annotated manuscript attached.

I hope these suggestions will improve the quality of the paper.

Best regards,

Dear reviewer, thanks for your time and expertise.
Your considerations helped to improved a lot this manuscript. We evalluated and answered them point by point below and I hope it is better presented in the current version.. After all corrections we sent the manuscript to review the english

Line 3: from the
Answer: done as suggested
Line 9: add a semi-colon (;) after the e-mail address
Answer: done as suggested

Line 17: use a common word for demersal here
Answer: word excluded

Line 18: , and can

(Or: start new sentence here: It can...)

Answer: Rewrited considering both reviewers

Line 19: rephrase
Answer: done as suggested

Line 22: For the common reader (general public, cf. ssimple summary), this is very difficult to understand: what does 'still mostly in some cases' means? Better to rephrase
Answer: We tried to elucidate

Line 23: in the
Answer: done as suggested

Line 24: Existing word?
Answer: corrected

Line 29: northeastern to southern

Answer: done as suggested

Line 31: highlighted
Answer: Rewrited

Line 34: from?
Answer: done as suggested

Line 38: Use a capital letter here
Answer: done as suggested

Line 41: Rephrase for easier reading? Or add a colon (:)? (para)symphyseal teeth

Answer: done as suggested.

Line 46: , and can
Answer: done as suggested

Line 49: The Bluntnose
Answer: done as suggested

Line 59: highlighted
Answer: done as suggested

Line 61: highlighted
Answer: done as suggested

Line 62: Do you have more references to make your case ‘stronger’, or is it just a single reference that has this hiatus?
Answer: Inserted another.

Line 99: Prefer not to use the word butcher, or butchery. Better is fishmonger
Answer: done as suggested

Line 100: , and a video...
Answer: done as suggested

Line 101: Better to rephrase
Answer: Rewrited

Line 113: highlighted
Answer: Accession added

Line 124-126: Delete
Answer: done as suggested

Line 127: With a capital letter
Answer: done as suggested

Line 130: Use a line above this title
Answer: This is the journal instructions

Line 132: I would prefer top ut the Figure first, and the table second
Answer: Actually the table should have been called in the previous paragraph, so it should come first

Line 134: I guess depth? Please specify.

Answer: done as suggested

Line 135: highlighted
Answer: done as suggested

Line 136: highlighted
Answer: done as suggested

Line 137: I would suggest to give the Figure first, and the table second. Check lay-out. table is now split over two pages, and caption is above (not below) table highlighted
Answer: See above. About the layout, it will be corrected later. The caption of table according to the jornal is above of the table.

Line 164: Caption is difficult to read. Make better use of the puntuation marks, or put letters a-b-c-d after the descriptions? highlighted
Answer: We tried to elucidate

Line 164: fisherman
Answer: It was the fishmonger, we corrected

Line 166: write in full?
Answer: done as suggested

Line 169: in 2005
Answer: done as suggested

Line 172: at
Answer: done as suggested

Line 174: I would suggest to treat the video as a reference (Author, year), and put the link in the list of references (+ accesion date).

Answer: We decided to present in the Supplementary Material added

Line 178: highlighted
Answer: done as suggested before

Line 178: a video
Answer: done as suggested

Line 179: same comment
Answer: Presented in Supplementary material

Line 182: 2
Answer: done as suggested

Line 194: Hexanchus spp.
Answer: done as suggested

Line 196: Illustrations after...?
Answer: We do not understand the question

Line Lines 198-201: Delete
Answer: done as suggested

Line 212: characteristic for
Answer: We used typical

Line 213: fishermen avoids
Answer: done as suggested

Line 213: highlighted
Answer: done as suggested

Line 258: This might be the result of sexual seggregation as recently suggested by Assemat et al. (2022) - Exploring diet shifts and ecology in modern sharks using calcium isotopes and trace metal records of their teeth - case Hexanchus griseus (Journal Fish Biology) https://doi.org/10.1111/jfb.15211 I would suggest to add this reference in line 267 when dealing with suggested research on the spatial ecology of the species.

Answer: Thanks so much! I did not know this reference.

Line 267: Suggestion to add Assemat et al., 2022 as extra reference
Answer: done as suggested

Line 274: In fact, the short summary, the abstract, the discussion and the conclusions all say the same. Some sentences are repeated several times throughout the manuscript. This is somewhat annnoying, especially in a short 'new record' paper as this. Are the conclusions needed? Or perhaps you can try to be more complement between these four sections to avoid repetition of the same sentences?

Answer: We rewrited simplifying

Line 283: delete
Answer: done as suggested

Line 285: delete
Answer: done as suggested

Line 285: a or the vídeo

Answer: done as suggested

Line 286: delete
Answer: done as suggested

Line 288: highlighted
Answer: done as suggested

Line 292: Please check/add name of series (i.e. FAO Fisheries Synopsis 125), and, if needed, name and location of publisher (city and country).

Answer: done as suggested

Line 295: species name in italics
Answer: done as suggested

Line 295: General remark for all journal references: please check/use dots to abbreviate journal names: e.g., Reg. Stud. Mar. Sci.

Answer: This is the journal instructions. Without dot

Line 298: General remark for all references: please be consistent in the presentation of DOI's, sometimes you use https://dx.doi.org/... elsewhere you use Doi: ....

Doi's are missing for many papers that have one

Answer: done as suggested

Line 299: Add locality of publisher, and total number of pages.

Answer: done as suggested

Line 300: 'South' should read 'Southern'

Answer: This is the article title. See https://www.ajol.info/index.php/ajms/article/view/33401

Line 301: highlighted
Answer: This is the journal instructions. Without dot

Line 303: 'hexanchoid' should read 'hexanchid'

Answer: This is the article title. See https://www.researchgate.net/profile/David-Ebert-7/publication/35714367_Aspects_of_the_life_history_of_California's_two_cowshark_species_Notorynchus_cepedianus_and_Hexanchus_griseus/links/556e73ce08aeccd7773f6ed4/Aspects-of-the-life-history-of-Californias-two-cowshark-species-Notorynchus-cepedianus-and-Hexanchus-griseus.pdf

Line 304: 'Ichthyology' should read 'Ichthyological'

Answer: done as suggested

Line 306: delete a space here
Answer: done as suggested

Line 308: highlighted
Answer: This is the journal instructions. Without dot

Line 310: highlighted
Answer: This is the journal instructions. Without dot

Line 313: Is 6th author in the original publication (not 4th). Please correct chronology of authors.

Answer: done as suggested

Line 319: highlighted
Answer: This is the journal instructions. Without dot

Line 320-321: highlighted
Answer: This is the journal instructions. Without dot

Line 322: Add a hyphen here
Answer: done as suggested

Line 325: highlighted
Answer: This is the journal instructions. Without dot

Line 327: 'Mentjies' should read 'Meintjes'

Answer: done as suggested

Line 328: See Doi remark above
Answer: done as suggested

Line 329: highlighted
Answer: This is the journal instructions. Without dot

Line 331: highlighted
Answer: done as suggested

Line 332: highlighted
Answer: This is the journal instructions. Without dot

Line 334: Add last page number
Answer: done as suggested

Line 336: highlighted
Answer: done as suggested

Line 341: highlighted
Answer: done as suggested

Line 348: Complete the reference, page numbers.
Answer: done as suggested

Line 349: Please complete the reference: name and location of publisher, number of pages...

Answer: done as suggested

Line 350: Please complete the reference: name and location of publisher, number of pages...

Answer: done as suggested

Line 352: highlighted
Answer: This is the journal instructions. Without dot

Line 353: Complete the reference please. Add editors etc.

Answer: done as suggested

Line 355-356: highlighted
Answer: This is the journal instructions. Without dot

Line 357: 'Walker' should read 'Waller'.

Answer: done as suggested

Line 357: highlighted
Answer: This is the journal instructions. Without dot

Reviewer 2 Report

The paper is quite repetitive and not written in a way to make it interesting when I think it could be. The methods are extremely vague and unclear. I realize it is a review of historical records mostly, but more details on how those prior records were collected and confirmed is needed as well as for the more recent work. It is also strange that a genetic component was included for what appears to be a single sample. I think it is great that a genetic confirmation was performed but the rest of that analysis seems unnecessary and takes away from the premise of the paper, which appears to be to highlight the occurrences of H. griseus off of the northeastern and central coasts of Brazil, which are missing from the IUCN distribution database. Although it seems other records in that area are in prior literature, yet excluded from IUCN's red list site for this species for some reason. It would be helpful if these other occurrences in the literature are included on your map and cited appropriately. I am not sure if you could get the rights to publish the IUCN distribution map to put alongside your records but that would make a nice visual comparison and show the "expansion" nicely. Attached in the pdf file are some additional and more detailed comments and edits on the article. 

Author Response

Reviewer 2

The paper is quite repetitive and not written in a way to make it interesting when I think it could be. The methods are extremely vague and unclear. I realize it is a review of historical records mostly, but more details on how those prior records were collected and confirmed is needed as well as for the more recent work. It is also strange that a genetic component was included for what appears to be a single sample. I think it is great that a genetic confirmation was performed but the rest of that analysis seems unnecessary and takes away from the premise of the paper, which appears to be to highlight the occurrences of H. griseus off of the northeastern and central coasts of Brazil, which are missing from the IUCN distribution database. Although it seems other records in that area are in prior literature, yet excluded from IUCN's red list site for this species for some reason. It would be helpful if these other occurrences in the literature are included on your map and cited appropriately. I am not sure if you could get the rights to publish the IUCN distribution map to put alongside your records but that would make a nice visual comparison and show the "expansion" nicely. Attached in the pdf file are some additional and more detailed comments and edits on the article. 

Answer: Dear reviewer, thanks for your time and expertise.
Your considerations helped to improve a lot this manuscript. We evalluated and answered them point by point below and I hope it is better presented in the current version..  After all corrections we sent the manuscript to review the english

Line 3: the

Answer: Done as suggested

Line 18: oceans

Answer: Done as suggested

Line 18: “yet patchy” is awkward. Rearrange the sentence

Answer: rewrited

Line 18: they

Answer: Rewrited considering both reviewers

Line 19: in- at /  sea-of (add in depths where they are found)

Answer: Rewrited considering both reviewers

Line 20: with

Answer: Done as suggested

Line 22: what do you mean by “mostly ignored”?

Answer: We tried to elucidate

Line 22: highlights

Answer: Done as suggested

Line 23: in the

Answer: Done as suggested

Line 23: particularly

Answer: Done as suggested

Line 24: establish

Answer: Done as suggested

Line 25: by

Answer: Done as suggested

Line 27: inhabits

Answer: Done as suggested

Line 28: of

Answer: Done as suggested

Line 31: see above

Answer: Rewrited

Line 44: What about the other species? Or explain why thus focuses on just this one

Answer: We just mentioned the diversity of the group and then focused on the species the manuscript investigated

Line 46: See all comments in abstract. Try not to use the same exact sentences in multiple places

Answer: Done as suggested

Line 49: bodies

Answer: Chenge by bodied

Line 50: delete “,” after males

Answer: Done as suggested

Line 54: delete “, and” and “are”

Answer: Done as suggested

Line 56: delete “it was detected”

Answer: Done as suggested

Line 56: insert was detected after” system”the

Answer: Done as suggested

Line 57: her

Answer: Done as suggested

Line 57: Why? Explain

Answer: Done as suggested

Line 61-62: see comment above

Answer: Done as suggested

Line 62: anthropenic

Answer: Done as suggested

Line 64: with

Answer: Done as suggested

Line 66: including

Answer: Done as suggested

Line 77:What about scientific expeditions?

Answer: We rewrited.

Line 78: sold?

Answer: Yes

Line 80: Explain more what this is

Answer: Done as suggested

Line 84: At what depth were the different numbers of hooks used?

Answer: every 50 m

Line 85: Explain more, does this means 53 different vessels or trips? Can you inlcude a map with these 53 locations? How long was the spacing between hooks and length of the longline?

Answer: We rewrite adding trips. We think insert locations of each fishing trip would polute the map and what we are trying to show. Also, we do not want to spoil these informations because the data are been used in a more embracing analysis

Line 91: What do you mean by fishery?

Answer: Rewrited

Line 93: Just one caught this way? Again this method needs further clarification.

Answer: Rewrited

Line 100: , after video

Answer: Done as suggested

Line 100: between?

Answer: Done as suggested

Line 101: delete “was made”

Answer: Done as suggested

Line 102:was

Answer: Done as suggested

Line 102: delete “y”

Answer: Done as suggested

Line 103: Only one sample was barcoded?

Answer: Rephrased

Line 106: What is mantle tissue? Muscle?

Answer: Corrected

Line 107:Explain

Answer: Done as suggested

Line 107: Write out all acronyms with their first use

Answer: Done as suggested

Line 113: ?

Answer: accession added

Line 120: Make sure acronyms are consistent

Answer: Done as suggested

Lines 124-126: Delete

Answer: Done as suggested

Line 127: N .... D

Answer: Done as suggested

Line 128: the

Answer: Done as suggested

Line 131: Which expeditions? Where? When?

Answer: Rewrited

Line 133: no verb in this sentence

Answer: Verb inserted

Line 134: depths?

Answer: Done as suggested

Line 136: Were these surveys only done in late 1998?

Answer: explained

Line 140: keep order the same as prior

Answer: We prefer to present the table in chronological order

Map: include names to go wih the state acronyms

Answer: Included in figure legends

Line 145: What are the dots? Include a legend

Answer: We erased the dots. It was confusing. We also included the previous records

Figure 2:What do these acronyms refer to?

Answer: Elucidated. Hg, Hexanchus griseus; Eb, Echinorhinus brucus.

Line 167: Still not sure what do you mean by fishery

Answer: Rewrited

Line 191: confirmed one specimen as H. griseus?

Answer: Clarified in the text

Line 198-201: Delete

Answer: Done as suggested

Line 208: habitat?

Answer: Yes. thanks

Line 209-211: What do you mean by poor taxonomic resolution?  Is there a lack of records or just that it isn't caught often?

Answer: Rewrited.

Line 212: typical

Answer: Done as suggested

Line 213: fishers

Answer: Done as suggested

Line 218: What are congresso abstracts?

Answer: changed . conference abstracts

Line 234: Why are they presumed different?

Answer: It was confusing. Rewrited

Line 240: delete “recorded [4]”

Answer: Done as suggested

Line 252: See Daly-engel et al. 2018 Ressurection of the sixgill shark Hexanchus vitulus Springer & Walker, 1969 (Hexanchiformes, Hexanchidae), with comments on its distribution in the northwest Atlntic Ocean

Answer: Thanks

Line 262: This is confusing. What do you mean high estimated size of adults and why would this make them absent from catches? Same with a refusal to ship large specimens?

Answer: We tried to elucidate and explain
